# Relative Phase Equivariant Deep Neural Systems for Physical Layer Communications

**Arwin Gansekoele**                                                                    *awg@cwi.nl*
*Stochastics Department*
*Centrum Wiskunde & Informatica, Amsterdam*

**Sandjai Bhulai**                                                                      *s.bhulai@vu.nl*
*Department of Mathematics*
*Vrije Universiteit Amsterdam*

**Mark Hoogendoorn**                                                          *m.hoogendoorn@vu.nl*
*Department of Computer Science*
*Vrije Universiteit Amsterdam*

**Rob van der Mei**                                                      *r.d.van.der.mei@cwi.nl*
*Stochastics Department*
*Centrum Wiskunde & Informatica, Amsterdam*

**Reviewed on OpenReview:** *https: // openreview. net/ forum? id= vttqWoSJIW*

## Abstract

In the era of telecommunications, the increasing demand for complex and specialized communication systems has led to a focus on improving physical layer communications. Artificial intelligence (AI) has emerged as a promising solution avenue for doing so. Deep neural receivers have already shown significant promise in improving the performance of communications systems. However, a major challenge lies in developing deep neural receivers that match the energy efficiency and speed of traditional receivers. This work investigates the incorporation of inductive biases in the physical layer using group-equivariant deep learning to improve the parameter efficiency of deep neural receivers. We do so by constructing a deep neural receiver that is equivariant with respect to the phase of arrival. We show that the inclusion of relative phase equivariance significantly reduces the error rate of deep neural receivers at similar model sizes. Thus, we show the potential of group-equivariant deep learning in the domain of physical layer communications.

## 1 Introduction

In the modern era, our world has become increasingly interconnected, with large amounts of information flowing seamlessly across continents. Telecommunications has been a key enabler in this transition to the era of information. However, as connectivity requirements become more extreme, the need for more specialized software-based networking systems increases. An important component in these developments is the physical layer of the networking stack. The physical layer manages the electrical and mechanical components of the transmission process, where the information is generally treated as some arbitrary sequence of bits to ensure compatibility with the rest of the stack. The modularity of the networking approach has allowed the rapid adaptation and development of telecommunications systems. The concept of a software-defined radio (SDR) could be a solution to many of the demands for increasing flexibility and speed. An SDR performs networking on a software level, as opposed to a hardware level, giving a high degree of flexibility at the cost of an increase in compute.

Given the recent advancements in AI as well, it becomes interesting to look into its potential for physical layer communications (Hoydis et al., 2021). The potential of AI for 6G is often considered to lie in its data-driven nature. When a signal with data is transmitted, it often arrives at a receiver distorted, requiring multiple modules to correct the errors and retrieve the original data. One type of distortion is called *fading*, which requires known pilots on the receiver side to perform data-driven channel correction. Korpi et al. (2021) proposed to replace a receiver entirely with a single deep neural network (DNN). They named this receiver DeepRx and demonstrated that replacing the entire receiver with a DNN resulted in significant performance gains in correcting fading channels.

Although this work showed the potential of DNNs in an unconstrained environment, a major problem remains that receivers have to be extremely fast and energy efficient. Although over-the-air DNN receivers are known to be feasible, they lack the throughput and energy efficiency to be viable on a large scale. As such, an important challenge is to make DNN models that are as efficient as possible without major reductions in performance. An approach to improve the efficiency of neural networks is to embed inductive biases into the network. Deep neural receivers commonly operate in complex baseband, which is a complex representation commonly used in telecommunications. When considering an orthogonal frequency-division multiplexing (OFDM) system, they can be further expanded into time-frequency signals. DeepRx has already implemented a form of inductive bias through its convolutional architecture. It performs convolutions across the time and frequency components, resulting in translational equivariance for both components.

To identify further inductive biases, we focus on the phase of the signal. Most digital modulation schemes use some form of amplitude and phase shifting to encode information. Information is encoded by adjusting the phase and amplitude at set sampling points. When receiving a transmission, the initial phase can vary depending on, e.g., the distance between sender and transmitter and the angles of the transmit and receive antennas. Unlike the relative phase shifts between sampling points, the initial phase offset of the signal does not affect the underlying data. However, synchronizing the initial phase offset of a signal is important to get the correct data sequence back. In principle, the initial phase of a signal is relative as it depends entirely on the moments chosen to measure the signal.

Without incorporating this knowledge, a deep receiver is unaware of the phenomenon of a relative phase before seeing any data. Deep receivers generally learn to correct for this phenomenon by observing many instances of fading channels. If the training data is sufficiently diverse, the DNN can allocate parameter sets that identify the absolute phase offset in the underlying signal. An interesting question is whether a model that is aware of this relative initial phase is more capable with fewer parameters than a model that is not. Here, we define a model that is aware of the relative phase as a model that behaves predictably under changes in the initial phase.

We propose to tackle this problem using the concept of group equivariance. Group theory provides a convenient mathematical framework to model symmetries, for example. In work by Cohen & Welling (2016), a general approach was proposed to encode discrete group equivariance into neural networks. In line with this field of research, we propose a framework for the construction of deep neural receivers for physical-layer communications that behave predictably under differences in the initial phase of the signal. In doing so, we found the need for element-wise equivariant solutions on regular grids of complex values, and propose a first approach for incorporating these in a DNN. The contribution of this paper is threefold. [1]

1. We identify and construct the components necessary to achieve relative initial phase equivariance for deep neural receivers.

2. We propose a modernized version of DeepRx (Korpi et al. (2021)) that incorporates these group equivariant components, alongside other recent improvements to the neural architecture.

3. We validate our approach empirically and demonstrate that including relative phase equivariance achieves a better trade-off between the number of parameters and the performance of the model.

---

[1]The code is available under: https://github.com/awgansekoele/relative-phase-equivariant-deep-neural-systems.git

## 2 Related Works

**Deep neural receivers.**  The first important domain to which we contribute is the field of deep neural receiver design.  Yue Hao et al. (2018); Xuanxuan Gao et al. (2018) were some of the preliminary works on performing the tasks of channel estimation, equalization, and symbol demodulation with a single deep neural network. These works were among the first to highlight the potential of deep learning in the context of receiver design.  Neev Samuel et al. (2019) extended this work to the MIMO setting and similarly achieved strong results.  Honkala et al. (2021) proposed DeepRx, one of the deep receivers based on modern CNN design choices along with considerations specifically for physical layer communications.  They showed that with a strong architecture, system performance can approach the theoretical limit in certain settings.  Korpi et al. (2021) extended DeepRx to a MIMO setting.  Cammerer et al. (2023) proposed using a GNN to better incorporate the interaction between users in a MIMO setting for a receiver such as DeepRx.  The MIMO users can vary, highlighting the need for a flexible solution.  Raviv & Shlezinger (2023) proposed a set of data augmentation strategies to enhance the training of deep receivers.  Huttunen et al. (2023) proposed DeepTx, which learns the sender part as opposed to the receiver part.  Raviv et al. (2023) investigated the use of meta-learning for rapid adaptation of deep receivers to new channels.  Pihlajasalo et al. (2023) looked into making deep OFDM receivers more efficient in terms of power use.  Gansekoele et al. (2024) proposed a neural receiver that can demodulate multiple constellation types simultaneously without having to adjust the model parameters.  To contribute to this literature, we are among the first to explicitly demonstrate that incorporating inductive biases into deep receivers can result in models that require significantly fewer parameters without loss of performance.

**Group equivariance for point clouds.**  The field of geometric deep learning or equivariant deep learning is a well-established (Bronstein et al., 2021).  One of the seminal works by Cohen & Welling (2016) introduced the concept of a G-equivariant network.  They propose a method to construct networks that are equivariant with respect to arbitrary discrete groups.  This paper provides the basis for our equivariant construction. We focus our work primarily on the domain of equivariant point-cloud learning, as we found that relative phase equivariance results in an element-wise equivariance.  One of the first DNNs that operated directly on point clouds is PointNet (Charles et al., 2017).  Although they included permutation invariance to point order, they did not investigate point rotation equivariance.  Thomas et al. (2018) proposed TensorField networks for general SO(3) point-cloud equivariance.  They use filters built from spherical harmonics to enforce rotation equivariance.  Esteves et al. (2019) proposed a network for discrete views of 3D point cloud data.  This approach differs substantially from TensorField networks in that they lift to the group as opposed to restraining the filters.  At a similar time, Li et al. (2019) proposed a simple architecture that is equivariant with respect to discrete rotation groups in 2D.  Building on these works, Chen et al. (2021) proposed a form of depthwise separable convolutions over groups for a more efficient pointcloud network. On the more fundamental side, Dym & Maron (2020) demonstrated the universality of G-equivariant point cloud networks.  Finally, Bokman et al. (2022) proposed a network for 2D point cloud recognition that can incorporate 2D-2D correspondences.  However, we still found that the needs of our work differ substantially because the sequence of 'points' is meaningful in the case of I/Q samples.  I/Q samples arrive at specific frequencies and times.  We found that the combination of operating on a grid with a requirement for element-wise equivariance poses unique challenges and solutions.  That is why we contribute to the field of G-equivariant deep learning by addressing (some of) the unique challenges of this new application domain.

## 3 Methods

To construct our model, we first discuss the preliminaries needed to capture relative phase equivariance appropriately.

### 3.1 Preliminaries

When thinking of absolute phase offsets, they most commonly result from *fading*, i.e., a signal changing in strength over time due to geographical positions, environmental effects, etc.  Fading can also occur when multiple instances of the signal arrive at the receiver out of phase.  Fading as an effect is often modeled as a

linear time-invariant (LTI) system $y = h * x + \mathbb{CN}(0, \sigma^2)$. Here, $x$ are the transmitted data carriers, $h$ the fading coefficients that are multiplied element-wise ($*$), $\mathbb{CN}$ a complex Gaussian distribution, $\sigma^2$ the noise variance, and $y$ a sequence of received I/Q samples. Modeling fading in such a manner gives us a clear way to evaluate the effect on the received signal. These fading effects are often modeled and generated using a channel model to perform system-level simulations. Some common channel models are Rician and Rayleigh fading. These fading models abstract away many of the underlying complexities of signal propagation by modeling fading as a random process. A Rician fading process can be modeled as

$$h = me^{i\theta} + s. \tag{1}$$

Here, $m$ is the magnitude of the direct path, and $s$ is the term representing the Rayleigh fading paths sampled as $s \sim \mathbb{CN}(0, \sigma^2)$. The term $\theta$ represents the initial phase of the line-of-sight (LOS) path in the Rician fading channel. The LOS path is the path with the most power during transmission. Often, the value $\theta$ is left at 0 to exclude it from the simulations. However, this introduces a disconnect between practical systems and simulation. The work of Özdogan et al. (2019) investigated the difference in performance between a system that knows the value of $\theta$ and one that is unaware of it. They found a difference between 2% and 50% of spectral efficiency depending on the benchmark system. Their results indicate that estimating the initial phase of the LOS component is important to correct for the underlying fading effects.

Frequently, channel models are used to simulate fading coefficients $h$ to train deep neural receivers. Many channel models exist, some realistic enough to validate practical systems directly (3GPP, 2006). Assume that we sample channel coefficients from a Rician fading model and sample a different initial phase $\theta \sim U(-\pi, \pi)$ to ensure proper simulation. A deep neural receiver can learn the characteristics of this channel by observing many different combinations of realizations of channel coefficients and symbols. For complicated channel models, it may take many realizations and many model updates before a good deep receiver is obtained. Different initial phases can result in additional learning challenges. To better demonstrate this, we plotted some common constellation types in Figure 5 in the appendix. One common characteristic of these constellations is that their shape is identical under 90-degree rotations. However, the bit labels the receiver now expects differ after the 90-degree rotation, resulting in bit errors. As such, it is essential that a deep receiver learns to identify the initial phase rotation in this case. Probably, the deep receiver has to build a form of redundancy to identify different cases of rotation and the features needed to identify these cases.

An interesting question is whether there is a more efficient approach. After all, if the deep receiver receives a signal in which only the initial phase differs, it should still be able to recover the original symbols. This is in part due to the fact that a phase shift in $y$ under the previously defined LTI is proportional to a phase change in $h$ under Equation 1. It is commonly known that a rotated complex Gaussian random variable is again a complex Gaussian random variable with the same parameters. It also does not matter whether we apply the rotation to $x$ or $h$, due to the linearity of the convolution operator. This is convenient, as a phase-shifted $h$ in practice can represent, e.g., the same fading channel but with the receiver starting closer together or farther apart. In an ideal world, a phase shift in $y$ should not affect the demodulation performance.

If some function $f$ gives the same results regardless of some set of transformations, the function of $f$ is often referred to as being *equivariant*. The concept of equivariance is commonly studied from the perspective of group theory. Groups are convenient when working with and reasoning about symmetries. A *group* is a set $S$ with an associated operation $\circ : S \times S \to S$ satisfying the following properties:

1. *The identity element*: There exists an element $e \in S$ such that for any $f \in S$ it holds that $e \circ f = f \circ e = f$.

2. *Inverses*: For any element $f \in S$, there exists an inverse $g \in S$ such that $f \circ g = e$.

3. *Associativity*: For any $f, g, h \in S$, it holds that $(f \circ g) \circ h = f \circ (g \circ h)$.

A definition for $f$ being invariant to $g$ can be given as

$$f(x) = f(g \circ x). \tag{2}$$

Furthermore, the definition of equivariance can be given as

$$g \circ f(x) = f(g \circ x). \tag{3}$$

### 3.2 $\mathbb{T}$-equivariance

Given this framework, we identify the desired type of equivariance. Given a sequence of received samples $y$, the deep receiver should have equivariant operations with respect to the global phase of $y$. Adjusting the global phase of $y$ is equivalent to multiplying all values in $y$ by some complex value with a magnitude equal to 1. This group is commonly referred to as the *circle group* and is well studied. It can be defined as

$$\mathbb{T} = \{z \in \mathbb{C} : |z| = 1\}. \tag{4}$$

To see that this group is appropriate for relative phase equivariance, it is easiest to look at the effect on the frequency domain. In the frequency domain, each component is represented by an amplitude and a phase. As the Fourier transform is linear, multiplying every element in the time domain is equal to multiplying every element in the frequency domain. Multiplications by a complex value with a magnitude of 1 alter only the angle of the complex value i.e. the phase of the frequency component. As such, a transformation by $\mathbb{T}$ results in an identical phase rotation of all components in the frequency domain.

We note that this group is isomorphic to the SO(2) group, which is the group of all 2D matrices with a determinant of 1. This relationship is important because the next question that arises is how we can encode the properties set in equation 2 and equation 3. The work of Cohen & Welling (2016) was among the first approaches to construct a neural network that is G-equivariant with respect to arbitrary discrete groups. Note that a neural network is nothing more than a composition of functions. If each function in this composition is G-equivariant, the whole network is G-equivariant. The combination of the circle group and their group equivariant work form the backbone of our framework.

It is important to note that a G-equivariant function no longer operates on the original space for some set of I/Q samples received $y$. Assuming an OFDM system, we operate on $\mathbb{C}^2$, which is a grid of frequency over time components. Thus, a set of $\mathbb{T}$-equivariant functions must operate on the semidirect product $\mathbb{C}^2 \rtimes \mathbb{T}$. A practical issue we face is that we cannot represent continuous sets within a computer. Fortunately, the group $\mathbb{T}$ has some convenient properties to overcome this issue. The group is Abelian, which means that the order in which a set of group actions is applied does not matter. Consequently, the circle group has countably infinite proper, closed, discrete subgroups. Each of these subgroups is a cyclic group defined by the $n$th roots of unity, where $n$ is the size of the subgroup. The $n$th roots of unity can be computed as

$$z_k = \exp\left(\frac{2\pi k i}{n}\right), \quad \text{for } k = 0, \dots n - 1. \tag{5}$$

The $n$th roots of unity divide the circle into equally large areas, with the first multiplication always being equal to 1 i.e. the identity element. Clearly, any multiplication of these numbers with each other results in another number from that group. Consequently, all of these sets are closed. The notion that these are subgroups of $\mathbb{T}$ is convenient, as building an equivariant network over discrete groups is much simpler.

### 3.3 Constructing the Equivariant Operators

We have identified all the components for the construction of a $\mathbb{T}$-equivariant neural receiver and can thus discuss the process of building one. Generally, the construction of a G-equivariant network follows a three-phase framework. These phases consist of a lifting convolution, group convolutions, and an invariance operator. First, a lifting convolution lifts the original input to the group space. In our case, we raise an OFDM signal from $\mathbb{C}^2$ to $\mathbb{C}^2 \rtimes C_n$ where $C_n$ refers to the cyclic group of order $n$. Lifting the input to the product space ensures that the model is unconstrained with respect to the functions it can learn. Given the roots of unity, the lifting operation can be defined as

$$y(k, f, t) = z_k \times x(f, t). \tag{6}$$

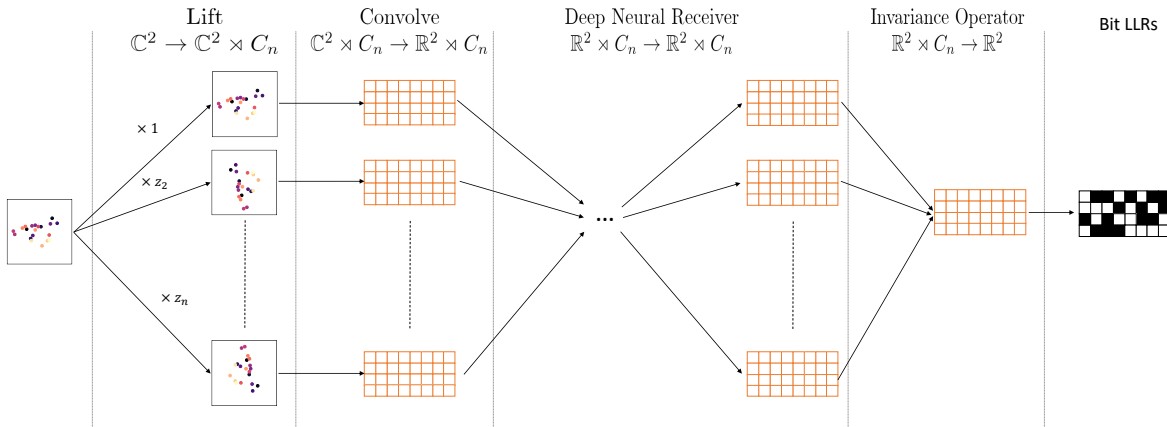

Figure 1: Demonstration of the flow of the equivariant model. A sequence of I/Q samples is lifted to $C_n$, pointwise convolved to a latent, transformed by the neural receiver, made invariant and finally transformed to bit LLRs.

Here, $x(f, t)$ refers to the input at frequency $f$ and timestep $t$. We note that most deep receivers do not actually operate on $\mathbb{C}^2$, however. Often, the choice is made to first transform the input from $\mathbb{C}^2$ to $\mathbb{R}^2$ to allow the use of recent developments in neural architecture design. We opted to mirror this approach and map our semidirect product $\mathbb{C}^2 \rtimes C_n$ to $\mathbb{R}^2 \rtimes C_n$. The first convolution thus translates the multiple complex-valued input stream to some higher dimensional representation without an explicit interpretation. In our neural network design, we opted to perform an element-wise linear map to perform this transformation, as it ensures equivariance while simplifying system design.

The consequence of this mapping is that the group is now represented as an ordered grid across the group dimension. Each element $k$ corresponds to a specific phase offset of the original signal. If we multiply the original input by some rotation from the $C_n$ group, the feature vectors of the original input would be the same, but their order would change. This result follows straightforwardly from the closed property of the group. Assume that we have a multiplication $\times z_j$ corresponding to a multiplication of some root of unity given the cyclic group $C_n$. As the group is Abelian, the order in which we perform this group operation does not matter, i.e., whether we apply it on the data first or directly on the vector of group transformations. The vector of group transformations can be written as a vector of multiplications by roots of unity as

$$\left[\times z_0, \ldots, \times z_{n-1}\right]. \tag{7}$$

As we know, this vector spans all of the elements of the group. Applying the same group operation again results in a vector containing all elements of the group as

$$\left[\times z_k, \ldots, \times z_{k+n-1}\right]. \tag{8}$$

A characteristic of the circle group is that given some $z_k$ with $k \geq n$, the value $z_k$ wraps around. Intuitively, a rotation by $3\pi$ gives the same result as a rotation by $\pi$. Interestingly, the permutation itself has a convenient ordering because of its cyclic nature. If we take the product of a group element with all group elements, the order is retained but shifted until the end of the circle is reached. At that point, it wraps around and the order resumes. Functionally, this means that the order of the feature maps is also cyclic.

To formalize this observation, consider a signal $s$ and a kernel $\psi$, where we assume both to be sequences operating in $C_n$. Take $C_n = \{e, z_1, z_2, \ldots, z_{n-1}\}$ as the cyclic group of order $n$ following previous definitions, with $e$ the identity element corresponding to multiplication by 1. Furthermore, define $L_{z_m}$ as the left action of $z_m$ and $f_\psi$ as a linear function operating on $s$ and defined using some kernel $\psi$. We then arrive at the following theorem:

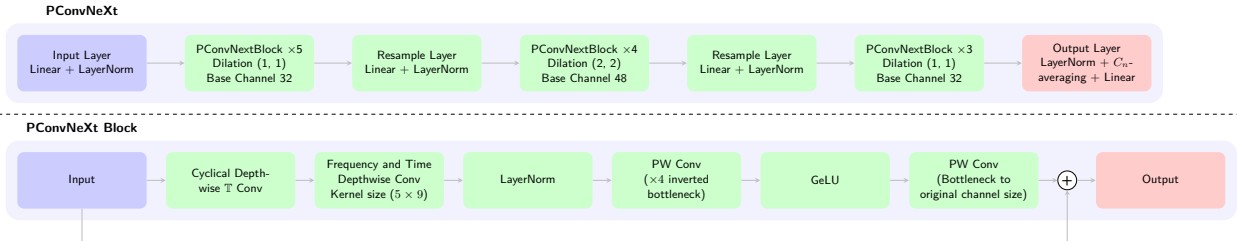

Figure 2: Schematic of the proposed ConvNeXt deep receiver with cyclical convolutions. Note that the cyclical convolutions become identity functions when considering $C_1$ (no equivariance).

**Theorem 1.** $L_{z_m}\left[f_\psi(s)\right](z_i) = f_\psi\left(L_{z_m}[s]\right)(z_i)$ *iff* $f_\psi$ *is a circular convolution.*

Here, $z_i$ is an arbitrary group element corresponding to $C_n$. The above theorem states that equivariance with respect to $C_n$ is not just guaranteed for circular convolutions but that any linear operation between two functions or sequences operating on $C_n$ necessarily has to be a circular convolution to ensure equivariance. This result demonstrates that using cyclic convolutions for equivariance is not only valid but also the most general and expressive solution possible. We note that most parts of the above result have been studied before in both mathematics and the geometric deep learning literature (Alkarni, 2001; Åhlander & Munthe-Kaas, 2005; Romero & Hoogendoorn, 2019). As such, we have opted to include the proof for the reader's convenience in the appendix under Section A.2.

We can thus learn parametrized functions that correlate locally relevant information across a variety of phase offsets. These cyclical convolutions can be implemented by using regular convolutions combined with circular padding. Circular padding implies padding the edges under the assumption that the data repeats cyclically. This padding can thus be implemented efficiently in all deep learning frameworks. Overall, this drastically simplifies the implementation of convolutions that are $C_n$-equivariant.

Finally, it is necessary to cast the results back to $\mathbb{R}^2$. Common aggregators over $C_n$, such as the mean and max operators, are sufficient to map from the product space $\mathbb{R}^2 \rtimes C_n$ to the desired latent space in $\mathbb{R}^2$. These are then translated to bit LLRs that can be used in a similar way as other neural receivers. Using such an aggregation method results in an output invariant with respect to $C_n$. To conclude, we now have all the components necessary to construct a $\mathbb{T}$-equivariant deep neural receiver. Note that this receiver does not depend on the choice of constellation. Any arbitrary constellation can be modeled with a $\mathbb{T}$-equivariant deep neural receiver. The components are visually depicted in Figure 1.

### 3.4 A Deep Phase Equivariant Receiver

Given the three operators that we discussed previously, we are now able to construct a neural network that is provably equivariant with respect to the phase of the original signal. To do so, we initially propose a more modern variant of DeepRx (Honkala et al., 2021). We base many of our design choices on the ConvNeXt architecture (Liu et al., 2022). In this work, the authors proposed various changes to common design choices in convolutional neural networks based on the success of the transformer architecture. As this architecture was built for image classification, we made multiple tweaks to make it suitable for OFDM signal processing. This resulted in the following overall design choices.

**Activation Functions and Normalization.** A common method to construct CNNs was to follow every convolution by both a normalization and an activation function. DeepRx also followed this design principle. As a normalization function such as LayerNorm is also nonlinear, ConvNeXt experimented with reducing the number of activation and normalization operations. They reduced the number of activations to one LayerNorm and one GeLU activation per block and found noticeable improvements. We mimic this design and adjust the activation functions for our ConvNeXt-based architecture.

**Inverted Bottleneck Block Structure.** Another design choice made to reduce the number of parameters without significantly reducing the expressive power of the architecture is the inverted bottleneck. DeepRx already included depthwise separable convolutions, which are based on the observation that a fully connected layer per kernel element is unnecessary. The inverted bottleneck design principle involves reducing the overall width of the model but temporarily increasing it within a block. This reduces the overall number of parameters, while still allowing the model to induce sparsity in higher dimensions.

**Increased kernel size.** The depthwise separable convolutions in DeepRx each had a kernel size of $3 \times 3$ with increasing amounts of dilation. One of the design concepts of the ConvNeXt architecture was to perform fewer convolutions but increase the size of the kernel, that is, one $7 \times 7$ instead of three $3 \times 3$. In a similar vein, we opt for a $5 \times 9$ filter, where 5 corresponds to the number of subcarriers and 9 to the number of OFDM symbols. The number of subcarriers is often much smaller than the number of symbols, hence why we decided on an imbalanced kernel. We found that this change also allowed us to reduce the amount of dilation.

The previous changes have no bearing on the equivariance operators. The equivariant components can be added to the architecture with some small changes. First, the projection layer is added at the beginning of the network. The network is positioned after the DFT module to extract the OFDM symbols, but the projection layer could also be applied before because of the linearity of the DFT. An important factor to note is that the input to DeepRx does not consist only of the signal. Both the pilot pattern and the least-squares estimates are included, as well, to improve convergence. These additional inputs must be managed appropriately to ensure that the network remains equivariant. Lifting the signal would consist of copying the signal $n$ times, where $n$ is the number of group elements, and rotating each by its corresponding root of unity. Naively copying and rotating the pilot patterns and least-squares estimates could cause problems. However, since the least-squares estimate is no more than a complex division of the received symbol by the pilot symbol, the neural network input remains valid if we multiply the least-squares estimate by the corresponding root of unity while leaving the pilot pattern unchanged. This is equivalent to rotating the input and then recomputing the least-squares estimates. We do not rotate the ground-truth pilot pattern, as these are predefined and should be independent of the initial phase of the signal.

For each of the $n$ rotations of the group $C_n$ that we include, we concatenate the rotated signal, the pilot pattern, and the rotated least-squares estimate. We then performed an element-wise linear transformation to lift each input to a higher-dimensional space. Afterward, as previously shown, the neural network operates on the semidirect product of $\mathbb{R}^2 \rtimes C_n$. Practically, this means that the input is now a batch of 3D elements. The three dimensions are the subcarriers, OFDM symbols, and group elements, respectively. By elementwise, we mean that we apply the same weights for every subcarrier, OFDM symbol, and cyclic group element.

Afterward, we insert a depthwise layer in every residual block that operates solely on the rotation dimensions. While a 3D-depthwise convolution that operates over all three dimensions is likely more expressive, common deep learning frameworks do not support GPU-accelerated implementations. By factorizing the 3D-depthwise convolution into 1D and 2D, we significantly improve the speed of the model at a small loss in expressivity. Finally, we implement a mean aggregation layer in the last feature map layer before mapping it to the bit detection layer. The joint architecture can be found in Figure 2.

## 4 Experimental Setup

For our experiments, we implemented the model in Figure 2. Unless otherwise mentioned, we maintain a specific structure for ease of consideration. The model consists of three stages, where stage 1, 2, and stage 3 consist of 5, 4, and 3 blocks, respectively. The word stage refers to a set of sequential blocks that share the same hyperparameter settings. Here, stages 1 and 3 have 32 channels, and stage 2 has 48 channels. We dilate the filters of stage 2 by a factor 2. This structure allows the network to widen the receptive field, after which it gets a simpler structure to correct.

To evaluate the effect of incorporating $C_n$-equivariance into a deep neural receiver, we implemented multiple sets of experiments. We used Sionna (Hoydis et al., 2022) to implement and evaluate our approach. Sionna provides TensorFlow implementations of physical layer components, as well as multiple 3GPP-compliant

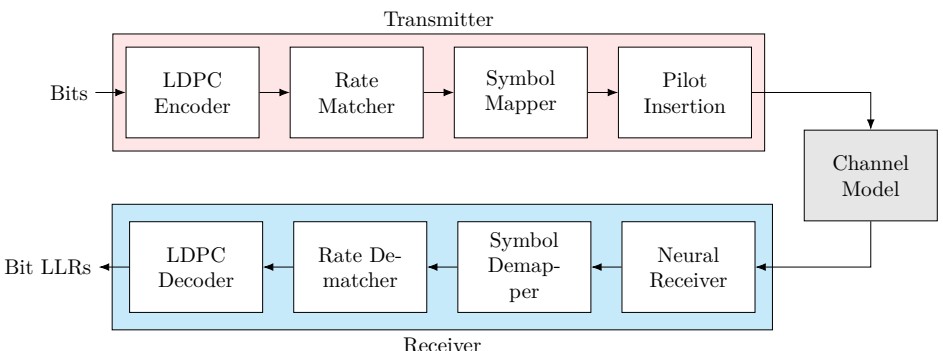

Figure 3: The OFDM sender/receiver pipeline that we used to evaluate our model.

channel models that are sufficiently realistic to allow conclusions on practical systems. We performed our experiments with a single-input multiple-output (SIMO) setup. This means that there is one transmit antenna and multiple receiver antennas. We chose this setup as it enables us to better isolate the capacity of the model without inducing additional complexity from multiple users. We opt for an OFDM system as it is the standard in 5G communications. We use the Urban Macro (UMa) channel model to generate channel coefficients and Gaussian noise to simulate receiver noise. We chose a UMa channel model as it is one of the most comprehensive channel models available designed for dense urban environments. We randomly sample valid user configurations for each set of channel coefficients. We evaluated all methods using the 3GPP-compliant TDL-A to C and CDL-A to E channel models. We chose multiple different channel models to ensure that our receiver works generally and not just for the UMa channel model. All channel models have one transmit antenna and two receive antennas. We show the overall communication pipeline in Figure 3.

We train each model for 150 epochs with the AdamW (Kingma & Ba, 2015; Loshchilov & Hutter, 2017) optimizer using an initial learning rate of $1e-3$. We use a stepwise scheduler that reduces the learning rate by a factor of 10 in the epochs 100 and 125. We found that this reduction helps stabilize the training. We found that this approach overall gives models sufficient time to converge. All experiments were performed on an A100 40GB GPU and repeated 10 times. After training our models, we evaluate the model based on the bit error rate (BER). We simulate until 500 batches of blocks were processed or 5,000 blocks with errors were detected. For all settings, we include both the results under least-squares estimation and when the channel is known (perfect csi). These serve as upper and lower bounds on the BER. If a neural receiver is worse than the LS estimate, we consider it non-functional since it receives the LS estimate as input. The code used to experiment is available at the following link.[2]

## 5   Results

**Main test bench.**   In Figure 4, we plot the BER curve of three methods for 16-QAM. We specifically focus on 16-QAM, as it is one of the most common constellation types. We included the constellations for visualization in Appendix A.1 in Figure 5. DeepRx (Honkala et al., 2021) serves as the baseline for a strong deep neural receiver, ConvNeXt is our approach without any group convolutions over a subgroup of $\mathbb{T}$, and ConvNeXt-$C_5$ adds the cyclical convolutions over $C_5$. We chose to use the subgroup $C_5$ for the equivariant approach unless otherwise mentioned. We chose $C_5$ as our results indicated during validation that this group achieves most of the performance gain without inflating the compute requirements too much. The results show that the ConvNeXt and DeepRx approaches have similar BER curves, despite the fact that ConvNeXt has less than a quarter of the parameters. Furthermore, ConvNeXt-$C_5$ achieves a BER curve that is tighter than both. This, again despite having less than a quarter of the parameters than the DeepRx model and slightly more than the ConvNeXt approach. We evaluated the mean BER of these curves and found that all differences were significant other than the difference between DeepRx and ConvNeXt. ConvNeXt-$C_5$

---

[2]Anonymous for now, will be made available in the definitive version.

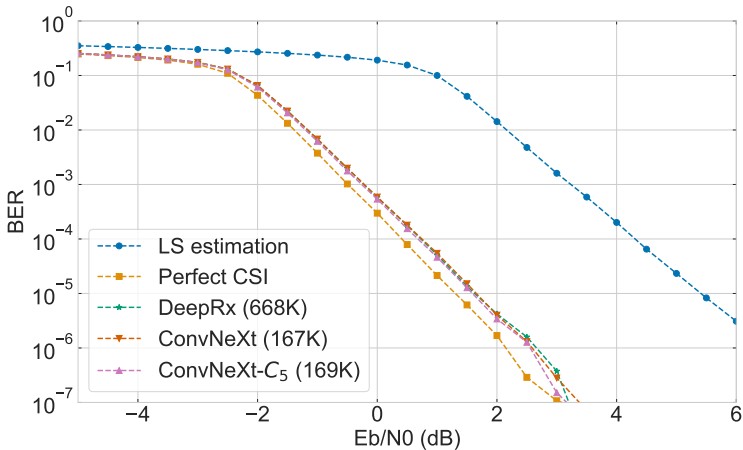

Figure 4: Comparison of the coded BER performance of DeepRx, ConvNeXt, and ConvNeXt-$C_5$ over 10 runs. The parameter counts are noted in the legend.

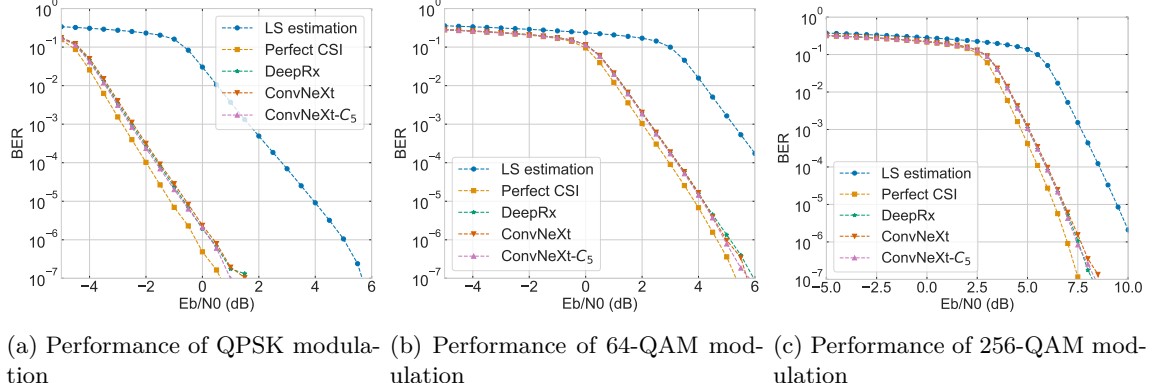

(a) Performance of QPSK modulation

(b) Performance of 64-QAM modulation

(c) Performance of 256-QAM modulation

Figure 5: BER curve comparison of the (a) QPSK, (b) 64-QAM and (c) 256-QAM constellation types.

therefore performs significantly better than ConvNeXt with a similar number of parameters and DeepRx with more than four times fewer parameters.

**Translation to other constellation types.** In Figure 5, we further compare ConvNext with ConvNeXt-$C_5$ for other common constellation types different from 16-QAM. Comparison to constellation types other than 16-QAM is interesting because of their different geometric properties. For instance, QPSK differs from the other constellation types in that it can be created by pure phase modulation. Interestingly, for QPSK, we see that the $C_5$ equivariant approach achieves the largest performance increase compared to the ConvNeXt model without $C_5$-equivariance. Although 64-QAM and 256-QAM both achieve a reduction in BER when including $C_5$-equivariance, we see that this reduction is more pronounced for QPSK. A possible explanation may lie in the lack of amplitude shifting that is present in the other constellation types we tested.

**Impact of model size.** We chose model sizes of 167K and 169K in previous experiments, as we found that these models present a good trade-off between parameter count and performance. However, the size of the model and the amount of computation needed for inference is often more important for deep receivers than small improvements in BER. If a receiver makes slightly more block errors but can process twice as many blocks in the same amount of time, the resulting bandwidth will be higher overall. That is why we opted to adjust the model size for both the ConvNeXt and ConvNeXt-$C_5$ receivers. We report the results of these experiments in Figure 6. In (a), we evaluated the adjustment of only the channel width. For example, the 618K ConvNeXt model is the result of doubling all channel widths in the original model in Figure 2. We

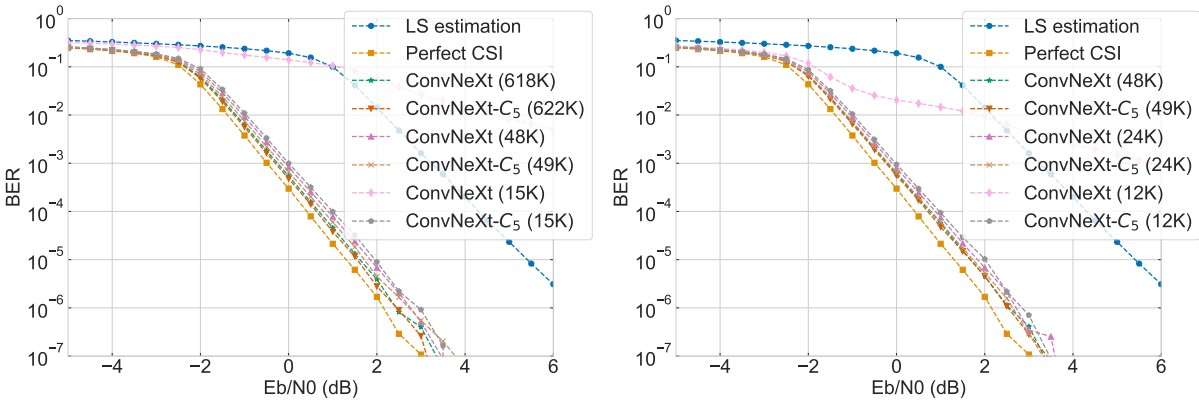

(a) Results of channel width adjustment experiments.

(b) Results of depth reduction experiments.

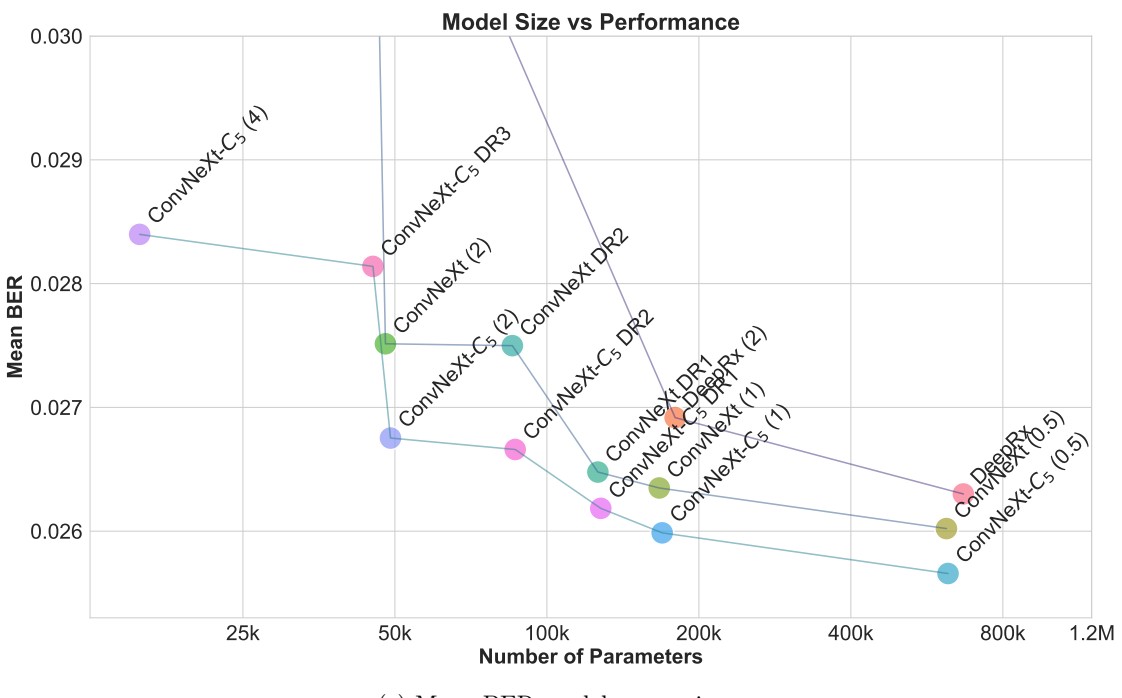

(c) Mean BER model comparison.

Figure 6: Experiments on model sizes: (a) impact of reducing model size, (b) impact of reducing model depth, (c) mean BER Pareto front illustrating trade-offs between model size and performance. DRn refers to all stages being reduced by $n$, where (n) refers to a division of all channels by $n$.

see that an increase in channel width results in an improvement in BER performance. We tested statistical significance in the mean BER and found that the difference between all models was significant.

However, we see anomalous behavior for the ConvNeXt (15K) model compared to the other models. At approximately 1.5 Eb/N0 (dB), the performance becomes worse than the LS estimate. We evaluated the runs and found that the model did not converge for some of them. We experimented with different sets of hyperparameters, but were unable to find one with better convergence properties. What is interesting is that the ConvNeXt-$C_5$ (15K) model managed to converge for all 10 runs, thus displaying better convergence properties at a similar number of parameters. Once we reduced the channel size further than the (15K) models, neither the ConvNeXt nor the ConvNeXt-$C_5$ models were able to converge.

Reducing the size of the model by changing the width of the channel maintains the same receptive field. To further evaluate the impact of parameter reduction, we opted to leave the channel width the same and reduce the number of blocks instead. We show the results in Figure 6b. Here, every reduction in parameters shows a reduction of 1 block per stage. For example, the 126K model has 4, 3, and 2 blocks in the three stages, as opposed to 5, 4, and 3. Again, we see that the larger models are more effective than the smaller ones. Interestingly, we again see divergence at the smallest model size. However, it occurs here for the 45K model when reducing the depth, as opposed to at 15K for reducing the channel width. This means that the model becomes unusable with a greater number of parameters than when width reduction is performed. The ConvNeXt model $C_5$ (45K) still manages to converge, demonstrating a higher capacity than the model without inductive bias.

Finally, we plotted the mean BER across the curve for all models compared to the size of the model in Figure 6c. There are multiple interesting observations to take from this figure. First, we see that both ConvNeXt (0.5) and ConvNeXt-$C_5$ (0.5) are (significantly) better in terms of mean BER at similar parameter sizes compared to the DeepRx model. What is also interesting is that ConvNeXt-$C_5$ (1) is similar in mean BER compared to ConvNeXt (0.5), despite having almost 4 times fewer parameters, with ConvNeXt-$C_5$ (1) achieving a (nonsignificantly) lower mean BER. As the inclusion of $\mathbb{T}$ equivariance induces an extra computation that scales linearly with the size of the discrete group, this result is highly interesting. We also included smaller versions of DeepRx, named DeepRx (2) and DeepRx (4). We see that the difference between DeepRx (2) and ConvNeXt (1) is larger than between DeepRx and ConvNeXt (0.5). This difference is even greater between ConvNeXt (2) and DeepRx (4). This demonstrates that ConvNeXt scales more gracefully in low-parameter regimes. Further, we see that depth reduction is a less efficient way to reduce the size of the deep neural receiver than channel width reduction, especially for smaller models. Finally, we see that ConvNeXt (4) is fully unable to converge, and we therefore chose to display it as a vertical line. Overall, we can see from the figure that the incorporation of the $C_5$-equivariance provides significant mean BER improvements at all the sizes of the models tested.

**Impact of group size.** We mentioned earlier that we use the group equivariant approach with $C_5$ as the discrete approximation of $\mathbb{T}$. We did so because we found that this approach provided a sweet spot between performance and the number of group elements that we needed to evaluate. To evaluate the impact of the choice of subgroup, we look at the following two variations. First, we tested the effect of different subgroups of $\mathbb{T}$ on the mean BER. Here, the mean BER refers to an average over the entire BER curve. We compared these with a ConvNeXt of similar size, denoted $C_1 - Base$, in Figure 7a. We can take multiple points from this comparison. First, we see that there is no performance improvement for $C_2$. While $C_2$ performs aggregation over two phase offsets, the kernel size is still 1. This means that no functions are learned across group elements. Second, we see that a small group such as $C_3$ is sufficient to achieve a significant performance increase. Interestingly, the next largest performance improvement comes from the transition of $C_3$ to $C_4$. Afterwards, no significant improvements are made in increasing the size of the subgroup. The 16-QAM constellation being identical under 90-degree rotations may explain why the BER reductions are maximal under $C_4$. Overall, our results indicate that the majority of the mean BER improvement is likely to come from the inclusion of the equivariance. Further, subgroup sizes larger than the possible phase ambiguities do not seem to make a significant improvement.

Secondly, we come back to the advantage of using cyclical convolutions. When performing convolutions over $\mathbb{T}$, the kernel size can be smaller than the size of the subgroup. This provides additional flexibility in the

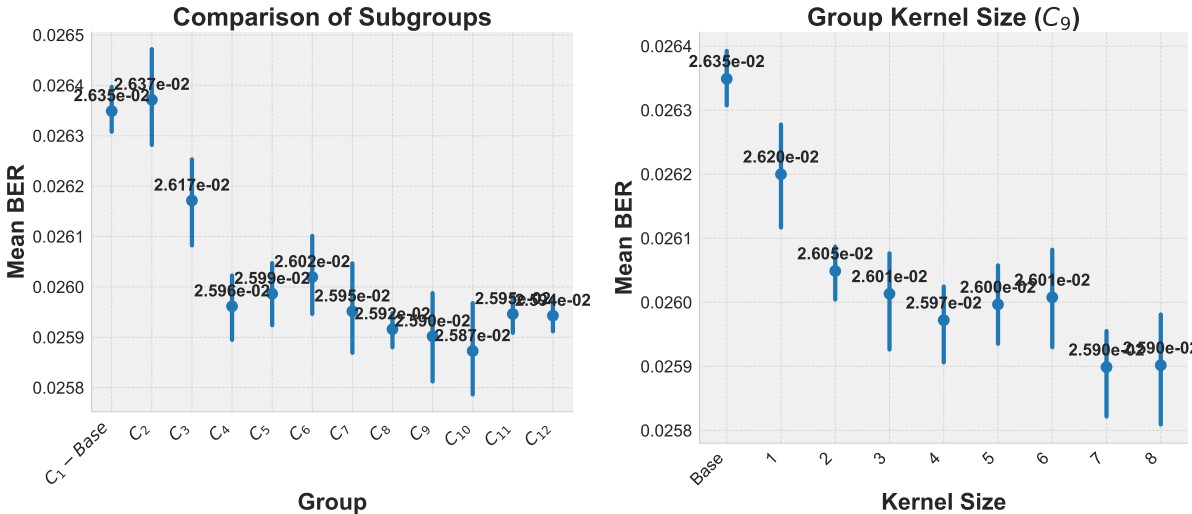

(a) Model performance across different group sizes.  (b) Model performance based on filter size variations.

Figure 7: Influence of group size and filter size on model performance: (a) Performance across varying group sizes, (b) Performance under $C_8$ based on different filter sizes. The bars contain the 95% confidence interval.

modeling since separate modeling choices can be made for the subgroup and the kernel size. To test this effect, we trained multiple ConvNeXt-$C_9$ receivers with different kernel sizes. We show their mean BER in Figure 7b. Note that Base again corresponds to a ConvNeXt without any group convolutions. What is interesting to note first is the result for a kernel size of 1. One of the largest reductions in BER when increasing the kernel size occurs when the group equivariance is added. The only difference between this model and a base ConvNeXt is the aggregation step over $C_9$. The group elements are, otherwise, never mixed throughout the model, and the parameter counts are identical for the base model and the kernel size 1 model. Also interesting is that we did not see this effect for $C_2$. This indicates that aggregation over sufficient group elements is necessary to achieve this effect. We again see the largest reduction in BER when moving from a kernel size of 1 to 2. Note that the effective kernel size of the overall model in this configuration is 13 due to the model containing 12 group convolutions. Interestingly, only for kernel sizes 7 and 8 do we find a significant difference in mean BER compared to a kernel size of 2. Thus, we again find that the $C_9$-equivariance already significantly improves the model without accessing patterns across elements of $C_9$. A kernel size of 2 that is sufficient for most of the mean BER reduction indicates that pattern recognition across elements of $C_9$ aids model performance.

**Impact of number of pilots.** To expand on whether $\mathbb{T}$ equivariance adds to the expressivity of the network, we evaluated it in a more challenging setting. In previous experiments, we placed the pilots in a Kronecker pattern at indices 2 and 11. This provides the network with information about the time-varying component of fading. As such, we opt to evaluate our approaches when we leave out the pilots at position 11. This scenario is no longer realistic, as performing an interpolation of the LS estimates between indices 2 and 11 is no longer possible. Thus, it makes time-varying fading impossible to correct over longer periods of time. However, its primary purpose is to provide an interesting stress test. We show the BER curves in Figure 8a. We see that both ConvNeXt (168K) and ConvNeXt-$C_5$ (169K) are able to achieve strong performance over the given channel models. Interesting is the divergent behavior of the ConvNeXt (48K) model; a behavior that we do not see for ConvNeXt-$C_5$ (49K). To get a better idea of what happens there, we plot the loss curves for both models across the runs in Figure 8. We observe that the ConvNeXt model struggles to converge in some runs. By averaging over these failed runs, the resulting BER curve becomes unusable. We attempted more hyperparameter tuning, but were unable to find a setting where the ConvNeXt model converges consistently. Thus, we find that the $C_5$-ConvNeXt model can operate at smaller sizes compared to ConvNeXt before having convergence problems.

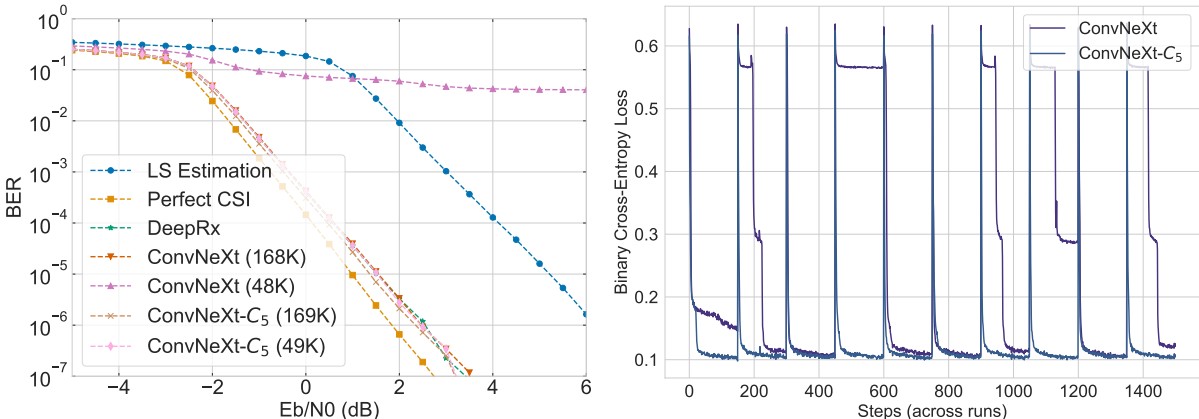

(a) Average performance across different pilot parameter sizes.

(b) Loss curve over the 10 runs for the ConvNeXt (48K) and ConvNeXt-$C_5$ (49K) models.

Figure 8: Analysis of using a single pilot: (a) average performance when only including a single pilot, (b) loss plot illustrating the convergence of the smaller single-pilot models.

## 6    Discussion & Conclusion

To recap, we proposed an element-wise $C_n$-equivariant DNN suitable for signals that benefit from phase equivariance. We applied our method to build a deep neural receiver that is equivariant with respect to variations in initial observation times and antenna angles. We implemented both a ConvNeXt and a $C_5$-equivariant ConvNeXt model, and tested them in a variety of settings. The core results can be summarized as follows.

1. For 16-QAM, the ConvNeXt approach performs similarly to DeepRx with four times fewer parameters. The group equivariant ConvNeXt-$C_5$ model outperforms both significantly in terms of mean BER.

2. The $C_5$-equivariant ConvNeXt networks significantly outperform base ConvNeXt models at all model sizes in terms of mean BER. It is also, on average, more capable in tiny model settings, due to better convergence properties.

3. The $\mathbb{T}$ equivariant approach performs relatively better for QPSK than for other constellation types.

4. We found that most performance benefits are achieved at $C_4$ and that larger subgroups of $\mathbb{T}$ do not provide significant benefits. We also found a small kernel size of 2 to be sufficient in a deep model under a group of $C_9$.

We found that a group equivariant approach performed well with a minimal increase in the number of parameters. An important aspect to discuss is whether the performance increase is the result of the group equivariance or other changes in the network. Some interesting implications arise from the ablation of the influence of the kernel size in this regard. It shows that computing the group elements and averaging at the end is sufficient for a major performance improvement. Furthermore, a kernel size of 2 is sufficient to achieve most of the BER reduction, which may imply that the core gain lies in the $\mathbb{T}$-equivariance.

Figure 5 also provides an interesting perspective on this question. These figures demonstrate that the performance increase depends on the constellation type as well. The larger the constellation, the less group equivariance improves the BER relatively. In the context of phase equivariance, this makes sense, as larger QAM constellations induce additional complexity that is not dependent on the initial phase.

To conclude, we proposed an $\mathbb{T}$-equivariant deep receiver and demonstrated that the $\mathbb{T}$-equivariance provides performance benefits at similar numbers of parameters. The main limitation of our approach is an increase in

computation that scales linearly with the size of the subgroup, which can make it impractical to implement in its current form. We note that for larger model sizes, we found that the $\mathbb{T}$-equivariance can provide similar performance at the same level of computation. We found that a $C_4$-equivariant model above a certain size achieves a similar BER to a $C_1$ model with 4x the number of parameters. We emphasize that we consider our core contributions to lie in the method of constructing a relative phase equivariant deep receiver and demonstrating that this property can improve the performance of deep neural receivers.

As our work is intended as preliminary work on the use of inductive biases to improve deep neural receiver design, future work could focus on the following aspects. The main aspect to look into is the improvement of compute times. The method scales linearly with the number of group elements, greatly limiting its practicality. Removing the need to evaluate all subgroup elements while retaining equivariance, for example, would be interesting. Research into other inductive biases of physical layer communications also holds potential. More practical limitations of our work include the lack of over-the-air validation and integration with traditional systems. Future work could look at both of these aspects. The approach would also enable direct generation of CSI, which is another interesting aspect to consider for backward compatibility with legacy systems.

### Broader Impact Statement

As our work is carried out on the physical layer of communications, which is an enabling technology, we do not expect a direct negative impact. The potential misuse of a better-performing communication system is outside the scope of this work. We do note the high energy requirements of the deep neural receivers in our experiments. The sum of performing these experiments costs multiple weeks of A100 computation. Addressing the energy requirements of deep neural receivers is an open problem.

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

# A   Appendix

## A.1   Additional Figures

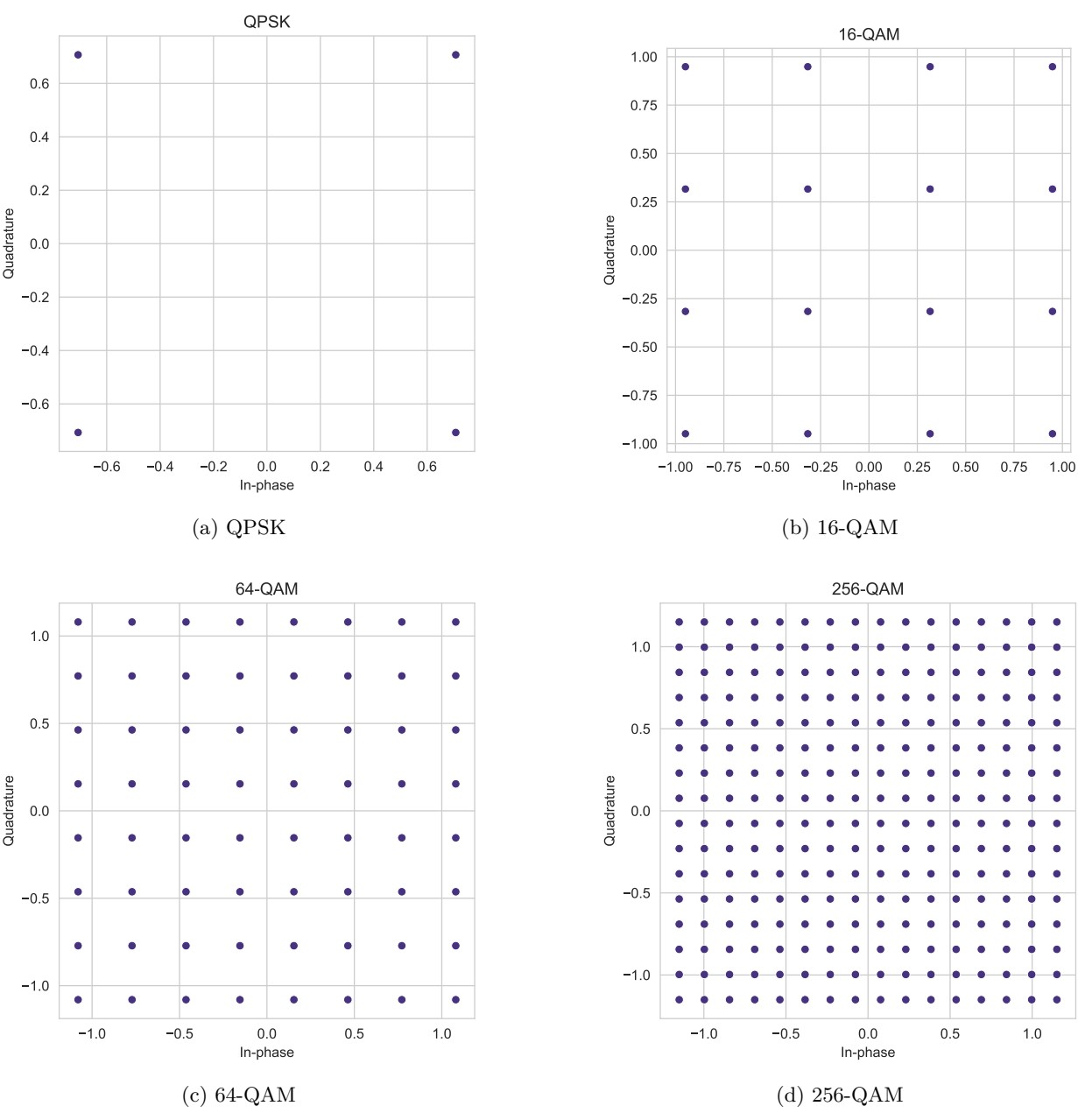

Figure 9: The QAM constellations used in our work.

## A.2 Circular Convolution Proof

As stated previously, we prove theorem 1 here.

*Proof.* As a reminder, we consider a function $f_\psi$ parametrized by kernel $\psi$ that operates on a signal $s$, where we assume $s$ and $\psi$ to be sequences over $C_n$. In addition, we take $C_n = \{e, z_1, z_2, \ldots, z_{n-1}\}$ as the cyclic permutation group of order $n$ with $e$ the identity element corresponding to multiplication by 1. In addition, $L_{z_m}$ is the left action of $z_m$. We will prove this theorem by showing both directions of the if and only if statement.

First, we prove that if $f_\psi$ is a circular convolution, then $L_{z_m}\left[f_\psi(s)\right](z_i) = f_\psi\left(L_{z_m}[s]\right)(z_i)$

As $f_\psi$ is a circular convolution, it follows that

$$f_\psi\left(s\right)\left(z_i\right) = \sum_{j=0}^{n-1} f(z_{j \mod n})\psi(z_{i-j \mod n}). \tag{9}$$

Let us take both sides of the equality.

$$L_{z_m}\left[f_\psi(s)\right](z_i) = f_\psi(s)(z_{i-m}) \tag{10}$$

$$= \sum_{j=0}^{n-1} s(z_{j \mod n})\psi(z_{i-m-j \mod n}). \tag{11}$$

We can now substitute $j' = j + m$. Since the convolution is circular and thus periodic, we can rearrange the elements of sum such that we again sum $j'$ from 0 to $n-1$, giving us the following

$$= \sum_{j'=0}^{n-1} s(z_{j'-m \mod n})\psi(z_{i-j' \mod n}) \tag{12}$$

$$= \sum_{j'=0}^{n-1} L_{z_m}\left[s(z_{j' \mod n})\right]\psi(z_{i-j' \mod n}) \tag{13}$$

$$= f_\psi(L_{z_m}[s])(z_i) \tag{14}$$

We thus find that if $f_\psi$ is a circular convolution, it follows that $L_{z_m}\left[f_\psi(s)\right](z_i) = f_\psi\left(L_{z_m}[s]\right)(z_i)$.

Second, we prove that if $L_{z_m}\left[f_\psi(s)\right](z_i) = f_\psi\left(L_{z_m}[s]\right)(z_i)$ then $f_\psi$ must be a circular convolution. We note that the function $f_\psi$ is a linear map and we can thus represent it as

$$f_\psi(s) = As. \tag{15}$$

Here, $A$ is some arbitrary linear matrix corresponding to the function $f_\psi$. We further note that we can represent the left action $L_{z_m}$ as a a permutation matrix $P$ that respects the cyclic permutation equivariance. This follows straightforwardly from the fact that $L_{z_m}$ operates on the cyclic permutation group $C_n$.

Thus, we can write the equivariance condition as $P(As) = A(Ps)$ for all $s$. Due to commutativity, it follows that $PA = AP$. As $P$ is a permutation matrix corresponding to the cyclic permutation, it follows that $P$ is a circulant matrix. As circulant matrices commute with each other, this means that $A$ must be a circulant matrix for $PA = AP$ to hold.

As $A$ is a circulant matrix, we can represent any arbitrary matrix $A$ as a circular convolution using elements of $\psi$.

$$A = \begin{bmatrix} \psi(e) & \psi(z_{n-1}) & \psi(z_{n-2}) & \ldots & \psi(z_1) \\ \psi(z_1) & \psi(e) & \psi(z_{n-1}) & \ldots & \psi(z_2) \\ \vdots & \vdots & \vdots & \ddots & \vdots \\ \psi(z_{n-1}) & \psi(z_{n-2}) & \psi(z_{n-3}) & \ldots & \psi(z_e) \end{bmatrix} \tag{16}$$

It thus follows that $f_\psi$ must be a circular convolution parametrized by $\psi$ for the equivariance condition to hold. In other words, if $L_{z_m}\left[f_\psi(s)\right](z_i) = f_\psi\left(L_{z_m}\left[s\right]\right)(z_i)$, then $f_\psi$ is necessarily a cyclic convolution.

Overall, we thus find that $L_{z_m}\left[f_\psi(s)\right](z_i) = f_\psi\left(L_{z_m}\left[s\right]\right)(z_i)$ iff $f_\psi$ is a circular convolution. $\qquad\square$

