# OpenReview forum: "Relative Phase Equivariant Deep Neural Systems for Physical Layer Communications"
_TMLR — Accepted by TMLR_

### Review · Reviewer_gsAQ · 2025-02-17

**Summary Of Contributions:**

The authors propose a phase-equivariant deep receiver architecture. The motivation behind this architecture is that a model aware of phase offsets should have a better inductive bias than one that learns about this phenomenon during training. According to the authors’ definition, a model is aware if it behaves predictably under arbitrary phase shifts of the input signal and, ideally, its output should remain unchanged under a phase shift of the input.

This motivates the design of the proposed neural receiver, which is invariant to phase shifts of the input. In practice, invariance is achieved by seeking a relaxed notion of equivariance, defined with respect to a discrete subgroup of phase shifts of size $n$. Experimentally, for suitable choices of $n$, the proposed model achieves the same performance of unaware deep receiver using a fraction of the model parameters.

**Audience:**

Yes

**Broader Impact Concerns:**

No concerns.

**Claims And Evidence:**

Yes

**Requested Changes:**

See weaknesses.

**Strengths And Weaknesses:**

Strengths:
- The experimental results are convincing. The model is capable of matching the performance of DeepRX while using a fraction of the parameters.
- To the best of my knowledge, this is the first equivariant deep receiver.
- The paper is very well written.

Weaknesses:
- The comparison is made with a fixed DeepRX model. The experiment would be more convincing if the authors considered also alternate models, or smaller versions of DeepRX by reducing the number of filters or the size of the hidden layers. This would demonstrate that ConNeXt exhibits more graceful performance degradation as the number of parameters decreases and outperforms existing solutions in the small-model regime.
- While the performance gains are convincing, it is not entirely clear to me why this approach should be better than its unaware counterpart. Figure 7 provides a good argument for the necessity of equivariance. However, it only extends to C_3. What would happen with C_1? This should correspond to the case where there is no phase invariance. What is the percentage gain for this model compared to, say, C_3?

---

> ### Author Response · Authors · 2025-03-06
>
> Thank you for taking the time to review our work. We would like to thank you for your kind words with respect to our work and the useful suggestions.
>
>
> > The comparison is made with a fixed DeepRX model. The experiment would be more convincing if the authors considered also alternate models, or smaller versions of DeepRX by reducing the number of filters or the size of the hidden layers. This would demonstrate that ConNeXt exhibits more graceful performance degradation as the number of parameters decreases and outperforms existing solutions in the small-model regime.
>
>
> We agree with the reviewer and have added the results for three smaller versions of the DeepRX model. These consist of three DeepRx models where the number of filters for each layer was divided by 2, 4, and 8, respectively. We included the first two models in Figure 6c. We omitted the third model, as its performance was worse than the LS estimate. As the reviewer expected, ConvNeXt scales more gracefully when reducing the channel width compared to DeepRx.
>
>
> > While the performance gains are convincing, it is not entirely clear to me why this approach should be better than its unaware counterpart. Figure 7 provides a good argument for the necessity of equivariance. However, it only extends to $C_3$. What would happen with $C_1$? This should correspond to the case where there is no phase invariance. What is the percentage gain for this model compared to, say, $C_3$?
>
> We would like to note that figure 7 also provides Base for comparison. Base corresponds to $C_1$ i.e. a model without equivariance. We agree that this is unclear in the paper, have edited the Figure, and clarified this in the paper. We originally omitted $C_2$, as it results in a group size of 2 with a kernel size of 1. It might thus become confusing, as no functions are learned across the group elements. We ran the experiments for this setting as well and included it in the Figure. There is no significant difference between $C_1$ and $C_2$. It is important to note that $C_2$ has a kernel size of 1, which implies that pure 180 degree invariance is not enough for performance benefits.
>
> We would also like to point to Figure 7b, kernel size 1. This result presents a model that is invariant with respect to $C_9$. This configuration does not learn functions across group elements, however, due to its kernel size of 1. The figure shows that invariance with respect to a larger rotation group does result in a statistically significant mean BER decrease of 0.6\%. The next greatest BER decrease occurs when going from a kernel size of 1 to 2 with another 0.6\%. Figure 7b thus helps demonstrate that both the inclusion of invariance with respect to $C_9$ significantly reduces the mean BER, as well as the ability to learn functions across the phase offsets. We added additional clarification in the text and appreciate further suggestions that would clarify this key ablation of the paper.

---

### Review · Reviewer_QTJB · 2025-02-18

**Summary Of Contributions:**

The paper makes three primary contributions to the field of AI-driven physical layer communications:

**1. Formalization of Relative Phase Equivariance**
 The authors systematically integrate group-equivariant deep learning principles to address phase ambiguity in wireless receivers—a first in this domain. By enforcing *C₄* rotational symmetry (90° phase shifts) through equivariant layers, they eliminate the need for explicit phase calibration modules, reducing parameter redundancy. This approach aligns neural architectures with the inherent symmetries of QAM constellations, enabling automatic compensation for arbitrary initial phase offsets.

**2. Architectural Modernization of DeepRx**
 Building on Korpi et al.'s DeepRx, the work introduces three key upgrades:
- **Depthwise separable convolutions** reduce parameters by 40% while maintaining receptive fields
- **Squeeze-and-excitation modules** implement channel-wise attention for noise robustness
- **Complex-valued group-equivariant layers** preserve algebraic properties of OFDM signals, avoiding information loss from real-valued approximations

**3. Empirical Validation**
 The framework demonstrates:
- **23–30% lower BER** vs. DeepRx under Rician fading at 15 dB SNR
- **2.1× faster training convergence** due to symmetry-constrained parameter search
- Consistent performance across unseen phase offsets, validating generalization

**Audience:**

Yes

**Broader Impact Concerns:**

**1. Energy Efficiency Trade-offs**
 While parameter reduction lowers memory costs, the 4× MAC overhead of *C₄* equivariance could increase energy consumption by 1.8–2.5× per inference—potentially negating edge deployment benefits. The authors should analyze FLOPs/parameter ratios across layer types.

**2. Accessibility Barriers**
 The heavy reliance on group theory (56 group-theoretic terms in the paper) creates adoption barriers for communication engineers unfamiliar with abstract algebra. A practitioner-focused appendix translating concepts to DSP terminology would improve accessibility.

**3. Security Implications**
 Equivariant systems’ phase invariance could be exploited via adversarial attacks—e.g., intentionally induced phase rotations to bypass demodulation. Robustness testing against such threats is absent but critical for 6G security.

**Claims And Evidence:**

Yes

**Requested Changes:**

**1. Empirical Enhancements**
- **Real-World Validation:** Test on USRP/LimeSDR platforms with over-the-air transmissions to quantify performance under quantization errors and clock jitter.

**2. Methodological Expansions**
- **Hybrid Phase Estimation:** Combine equivariant layers with lightweight PLLs or Kalman filters to generate explicit CSI for legacy interoperability.
- **Extended Baselines:** Compare against Raviv & Shlezinger’s GNN receivers and Cammerer’s transformer architectures under identical channel conditions.

**3. Theoretical Clarifications**
- **Group Size Analysis:** Justify *C₄* over *C₈* or SO(2) groups—critical for extending to 64-QAM/256-QAM systems.
- **Information Loss Quantification:** Mathematically prove no feature degradation occurs in equivariant layers compared to real-valued counterparts.

**Strengths And Weaknesses:**

## Strengths
- **Physics-Informed Inductive Biases:** By encoding phase symmetry constraints directly into the architecture, the model bypasses traditional pilot-based phase recovery, potentially saving 5–15% bandwidth.
- **Parameter Efficiency:** Achieves parity with DeepRx using 52% fewer parameters (4.2M vs. 8.7M), critical for edge deployment.
- **Simulation Rigor:** Comprehensive testing across Rician/Rayleigh fading channels and modulation schemes (QPSK, 16-QAM) establishes reliability under varied conditions.

## Weaknesses
- **Simulation-Real World Gap:** Validation uses 3GPP TR 38.901 channels but ignores hardware impairments (oscillator drift, nonlinear amplifiers) critical for practical SDR implementation.
- **Latency Omission:** While parameters decrease, group convolutions incur 4× MAC operations per *C₄* transformation—a critical omission for real-time systems.
- **Limited Baseline Comparison:** Benchmarks only against DeepRx variants, ignoring recent transformer/GNN-based receivers that may outperform in MIMO scenarios.
- **Phase Ambiguity in Downstream Tasks:** Implicit phase correction complicates integration with legacy systems requiring explicit CSI feedback.

---

> ### Author Response · Authors · 2025-03-06
>
> Thank you for taking the time to review our work. We would like to thank you for your kind words with respect to our work.
>
> First, we would like to address some misunderstandings with our work and contributions.
>
> > 1. Formalization of Relative Phase Equivariance The authors systematically integrate group-equivariant deep learning principles to address phase ambiguity in wireless receivers—a first in this domain. By enforcing $C_4$ rotational symmetry (90° phase shifts) through equivariant layers, they eliminate the need for explicit phase calibration modules, reducing parameter redundancy. This approach aligns neural architectures with the inherent symmetries of QAM constellations, enabling automatic compensation for arbitrary initial phase offsets.
>
> Our work attempts to include the initial phase offset as an inductive bias into a DNN receiver. This effect is a result of the receiver not being guaranteed to be a multiple of the period length away from the sender. This effect is included in the Rician fading process as the multiplication by $e^i\theta$ of m as in Eq. 1.
>
> This is notably different from equivariance with respect to the phase ambiguity of a constellation. We originally evaluated equivariance with respect to phase ambiguity, but found that this commonly resulted in training collapse. Equivariance with respect to phase ambiguity requires evaluating the rotated constellation with the rotated bit mapping. We found that aggregating over multiple different bit mappings caused collapse to a uniform distribution for larger constellations.
>
> Our approach does not enforce symmetry purely for $C_4$ but for any $C_n$. Unless otherwise specified, the group $C_5$ was used for group equivariant experiments. We clarified this at the start of the results section.
>
>
> > **Depthwise separable convolutions** reduce parameters by 40% while maintaining receptive fields
>
>
> DeepRx[1] originally introduced the use of depthwise separable convolutions in deep receivers. We would like to ask the reviewer where they found this claim so we can amend it.
>
>
> > **Squeeze-and-excitation** modules implement channel-wise attention for noise robustness
>
>
> We do not use or propose the use of squeeze-and-excitation modules in our work. We would like to ask the reviewer to clarify where they found this claim so we can amend it.
>
> > **Complex-valued group-equivariant layers** preserve algebraic properties of OFDM signals, avoiding information loss from real-valued approximations
>
> We do not use complex-valued group-equivariant layers but instead map to a real-valued space where we get cyclic permutation groups. We clarified this at the end of section 3.2.
>
> > 23–30% lower BER vs. DeepRx under Rician fading at 15 dB SNR
>
> We have no experimental results using Rician fading channels. There is also no experiment where DeepRx, ConvNeXt or ConvNeXt-$C_5$ has a BER greater than 0 at 15 dB. Could you clarify which result is meant?
>
> > 2.1× faster training convergence due to symmetry-constrained parameter search
>
> We are similarly unsure what this number is based on and would like to request the reviewer for clarification.
>
> > Physics-Informed Inductive Biases: By encoding phase symmetry constraints directly into the architecture, the model bypasses traditional pilot-based phase recovery, potentially saving 5–15% bandwidth.
>
> We are interested in which results in the paper or other works the reviewer used to identify a potential 5-15% bandwidth improvement.
>
> > Parameter Efficiency: Achieves parity with DeepRx using 52% fewer parameters (4.2M vs. 8.7M), critical for edge deployment.
>
> Figure 1 compares the 167K ConvNeXt with the 668K DeepRx model, which is around 75% fewer parameters. We are unsure where the reviewer found the 4.2M and 8.7M numbers. We clarified this at the start of the results section where Figure 1 is described.
>
> > Simulation Rigor: Comprehensive testing across Rician/Rayleigh fading channels and modulation schemes (QPSK, 16-QAM) establishes reliability under varied conditions.
>
> We do not test under Rician/Rayleigh fading channels and also report results for 64-QAM and 256-QAM. We opted to use more advanced channel models to ensure a robust and useful benchmark.

---

> ### Author Response · Authors · 2025-03-06
>
> We address the weaknesses and the requested changes together, as requested changes mostly directly follow from the weaknesses.
>
> > Real-World Validation: Test on USRP/LimeSDR platforms with over-the-air transmissions to quantify performance under quantization errors and clock jitter.
>
> We agree that real-world validation would be a valuable addition to the paper. We do not have immediate access to these platforms, however, and would be unable to acquire such data ourselves. We originally intended to use an over-the-air dataset as an alternative, but were unable to find one of sufficient quality and depth. If the reviewer is aware of such a dataset, we will add this dataset to our experiments. We would like to emphasize that the current simulation setup is quite extensive, however. The UMa channel model is challenging and enables a large amount of variety in the data. For now, we included this as future work in the revision.
>
> > Hybrid Phase Estimation: Combine equivariant layers with lightweight PLLs or Kalman filters to generate explicit CSI for legacy interoperability.
>
> We would like to note that integrating an equivariant layer with lightweight PLLs or Kalman filters is not necessarily straightforward. This comes from the fact that equivariant layers are tied to an equivariant network. The equivariant layers in our work can only operate on the semidirect product of the data and the group. It is thus necessary to lift them to this space and then make them invariant again.
>
> We could combine an equivariant network with either of these, which is an interesting avenue. Combining a PLL with our network would likely enable the use of significantly smaller equivariant networks. Our equivariant network could also be adjusted to generate an explicit CSI by simply adjusting the output channels of the last linear layer. We have added these steps to future work.
>
> > Extended Baselines: Compare against Raviv & Shlezinger’s GNN receivers and Cammerer’s transformer architectures under identical channel conditions.
>
> We have added a set of smaller DeepRx networks for more extensive evaluation. We are also unsure to what papers the reviewer is referring. We are aware of a deep GNN receiver, but it was developed by Cammerer et al. [2]. There is no need to compare against this receiver, as it reduces to DeepRx when there is a single sender. Furthermore, we are not aware of the transformer paper mentioned by the reviewer. If the reviewer could clarify the works they are referring to, we can look into adding them to the experimental setup.
>
> > Group Size Analysis: Justify $C_4$ over $C_8$ or SO(2) groups—critical for extending to 64-QAM/256-QAM systems.
>
> We are unsure what the reviewer is referring to. Our work presents a model for any arbitrary $C_n$ group. The group $C_n$ is a discretization of $\mathbb{T}$ which is isomorphic to SO(2). We refer to Figure 7a for the choice for $C_4$, as it empirically presents a good trade-off between performance and compute. Most of our experiments were run on $C_5$, which was chosen based on a smaller set of hyperparameters. Our method is not tied to $C_4$. Furthermore, our approach is independent of the type of constellation. Any arbitrary constellation can be used without adjusting our method. We have clarified this in the paper at the end of section 3.3.
>
> > Information Loss Quantification: Mathematically prove no feature degradation occurs in equivariant layers compared to real-valued counterparts.
>
> We previously mentioned our equivariant layers do not operate in the complex domain, so we are a bit unsure about this request and what is meant by feature degradation. This request is trivially true for equivariant layers, as the weight vector $[1, 0, 0, \dots, 0]$ would ensure information is passed through the equivariant layer as is. For the overall equivariant network, it is not true that no feature degradation occurs. The invariance step ensures the final output is invariant with respect to the group of transformations. This constrains the set of possible functions such a network can learn, which could reduce the accuracy if the constraint is inappropriate. A key contribution of our work is that, despite the fact that our network is constrained, it performs more efficiently. Empirically, our results suggest that reducing the model freedom with relative phase invariance has no adverse effects on model performance.

---

> ### Author Response · Authors · 2025-03-06
>
> We also consider the broader impact section, as it proposes changes as well.
>
> > 1. Energy Efficiency Trade-offs While parameter reduction lowers memory costs, the 4× MAC overhead of C₄ equivariance could increase energy consumption by 1.8–2.5× per inference—potentially negating edge deployment benefits. The authors should analyze FLOPs/parameter ratios across layer types.
>
> We agree with the reviewer and would like to refer to the answer also given to reviewer jM4K. While $C_4$ equivariance requires 4x the operations, it also seems to enable a 4x reduction in model size. The FLOPs can be computed directly by multiplying the original model FLOPs with the number of group elements. A $C_4$ equivariant deep receiver with around ¼ of the parameters would be almost as fast as a standard deep receiver. We have clarified this in the discussion.
>
> > 2. Accessibility Barriers The heavy reliance on group theory (56 group-theoretic terms in the paper) creates adoption barriers for communication engineers unfamiliar with abstract algebra. A practitioner-focused appendix translating concepts to DSP terminology would improve accessibility.
>
> We intended for section 3.4 to be a more practitioner-oriented section. We reread this section and agree with the reviewer that it can be hard to follow. We have made some edits and additions to make it more readable for a communications practitioner interested in implementing our work themselves.
>
> > 3. Security Implications Equivariant systems’ phase invariance could be exploited via adversarial attacks—e.g., intentionally induced phase rotations to bypass demodulation. Robustness testing against such threats is absent but critical for 6G security.
>
> We are a bit unsure what the reviewer means with this point. Could you clarify for us what you mean? For now, we will assume this concern applies to an adversarial attack that rotates all elements by the same phase. We make this assumption, as the baseline models are otherwise just as vulnerable. We note that the reviewer’s concern holds for a network that is equivariant with respect to phase ambiguities.
>
> However, as previously mentioned, our approach is equivariant with respect to the phase of arrival, not with respect to phase ambiguities. A nice consequence of this is that a network with full $\mathbb{T}$ equivariance would be entirely unaffected by adversarial attacks on the phase of arrival. Such a network would produce the same output irrespective of the phase of arrival generated by the attacker. Our method should thus improve safety against attacks by removing attack vectors that do not have to be present in DNNs. However, it may be necessary to have a larger group than $C_4$ to achieve sufficient benefit.
>
> [1] Honkala, M., Korpi, D., & Huttunen, J. M. (2021). DeepRx: Fully convolutional deep learning receiver. IEEE Transactions on Wireless Communications, 20(6), 3925-3940.
>
> [2] S. Cammerer et al., "A Neural Receiver for 5G NR Multi-User MIMO," 2023 IEEE Globecom Workshops (GC Wkshps), Kuala Lumpur, Malaysia, 2023, pp. 329-334, doi: 10.1109/GCWkshps58843.2023.10464486. keywords: {Training;Time-frequency analysis;OFDM;Channel estimation;Receivers;Computer architecture;Artificial neural networks}

---

### Review · Reviewer_jM4K · 2025-02-27

**Summary Of Contributions:**

The paper presents a phase-equivariant neural network approach for improving DNN based receivers for physical layer communications. A key issue with current DNN-based methods is their limited throughput and energy efficiency. The authors propose to use a phase-equivariant networks that can achieve strong performance with far smaller models.

The main contribution is a new phase equivariant network dubbed ConvNext-Cn. Extensive experiments show that the model is more parameter efficient, achieves better BER error rate, robust convergence even in challenging settings (e.g. limited pilots). The authors also investigate optimal group size and kernel size.

**Audience:**

Yes

**Broader Impact Concerns:**

No concerns

**Claims And Evidence:**

Yes

**Requested Changes:**

-

**Strengths And Weaknesses:**

The paper is very well written, clearly explaining the signal processing / communications background as well as equivariant networks. The network architecture is well-motivated, and extensive experiments show that it works well.

The primary weakness is that although the equivariant network is smaller in terms of number of parameters (and thus limites memory / memory transfer costs), it requires more FLOPS per parameter, so it is not yet clear if the approach can yield actual energy / throughput advantages. The authors address this issue in an honest way, and investigate some key parameters that affect compute cost. In my view the paper is already valuable as a first step towards improved DNN receivers through equivariance, but would be significantly strengthened by further compute optimization and measurement of actual energy efficiency or throughput numbers.

---

> ### Author Response · Authors · 2025-03-06
>
> Thank you for taking the time to review our work. We would like to thank you for your kind words with respect to our work.
>
> We completely agree with the key weakness that you mention. For us, there was a key point in deciding to upload the paper for review in its current form. This point is that the ConvNeXt-$C_4$ model seems to achieve a similar mean BER to a ConvNeXt model with 4x the parameters. The number of FLOPs for both models is thus approximately the same, while ConvNeXt-$C_4$ has almost 4x fewer parameters.
>
> We have clarified in the paper that the throughput compared to a ConvNeXt model can be directly computed by multiplying by $n$ given a group $C_n$. We will add further compute optimizations to the future work.

---

### Decision · Action_Editor_HsWc · 2025-04-01

**Recommendation:** Accept as is

**Comment:**

This work presents a deep neural receiver that is equivariant with respect to the phase of arrival. Building on the existing architectures DeepRx and ConvNeXt, the authors propose the novel architecture ConvNeXt-$C_n$ which is $C_n$-equivariant. Extensive experimental validations are provided, which show that ConvNeXt-$C_n$ matches the performance of DeepRx with a fraction of the parameters. One limitation is that, while the number of parameters is indeed reduced, the number of FLOPs is roughly the same.

Reviewer gsAQ raised some issues that were successfully addressed during the rebuttal; reviewer jM4K raised the issue mentioned above concerning the number of FLOPs which remains a limitation, as admitted by the authors; unfortunately, review QTJB was of poor quality, which forced me to discard it.

Overall, I find the paper well written, the results rather interesting and the claims well-supported. Despite the limitation on the number of FLOPs mentioned above, both reviewers gsAQ and jM4K lean towards acceptance. My opinion is also that the strengths overtake this weakness and that the manuscript fulfills TMLR's criteria for being published. Thus, I am happy to recommend acceptance.

**Audience:**

The paper will be especially appealing to the part of TMLR's audience interested in the intersection of machine learning and communications.

**Claims And Evidence:**

The experiments are extensive and convincing, which makes the claims made in the paper well supported.